# Validity and Repeatability Characteristics of a Non-Invasive, Infrared-Based Method Estimating Plasma Indocyanine Green Decay in Healthy Dogs

**DOI:** 10.3390/ani13223455

**Published:** 2023-11-09

**Authors:** Juhana Honkavaara, Agnieszka Grobelna, Flavia Restitutti, Ira Kallio-Kujala, Marja Raekallio, Thomas Spillmann

**Affiliations:** Faculty of Veterinary Medicine, University of Helsinki, FIN-00014 Helsinki, Finlandflavia.restitutti@ucd.ie (F.R.); ira.kallio-kujala@helsinki.fi (I.K.-K.); marja.raekallio@helsinki.fi (M.R.); thomas.spillmann@helsinki.fi (T.S.)

**Keywords:** indocyanine green, clearance, infrared, noninvasive, dog, medetomidine

## Abstract

**Simple Summary:**

Dogs are affected by various types and degrees of liver disease and dysfunction. Indocyanine green is an indicator dye cleared from the bloodstream exclusively by the liver. Hence the rate of its disappearance can potentially indicate liver dysfunction in dogs. We compared a novel, non-invasive, infrared-based method of detecting indocyanine green with a reference test that requires repeated blood samples. We completed both methods in parallel in both conscious and sedated dogs. Furthermore, we repeated the noninvasive test in conscious dogs to assess its repeatability. There did not appear to be a clear association between the two methods, possibly owing to signal quality issues with the non-invasive method. In addition, the repeatability of the noninvasive method was poor. We conclude that the noninvasive method performed poorly in healthy dogs, which should be considered when investigating dogs with clinical liver disease.

**Abstract:**

Plasma clearance of indocyanine green (ICG-CL) is an invasive method to evaluate liver dysfunction. We aimed to investigate the practicality of a noninvasive, transcutaneous, infrared-based method estimating the disappearance rate of indocyanine green (ICG-PDR). In a randomized, cross-over study, both ICG-CL and ICG-PDR were determined in eight healthy dogs while conscious and when sedated with medetomidine and medetomidine–vatinoxan. ICG-PDR was further repeated in six of the dogs to assess its repeatability. Differences were tested with repeated-measures analysis of variance and post hoc t-tests with Bonferroni corrections, while associations were evaluated by both Spearman and Pearson correlation analyses. Furthermore, repeatability was assessed by examining calculated coefficients of variation (CV). A significant decrease in ICG-CL was observed in dogs sedated with medetomidine, while no difference between conscious and sedated states was detected with ICG-PDR. Overall, correlations between ICG-CL and ICG-PDR were poor, as was the intrasubject repeatability of ICG-PDR in conscious dogs with CV consistently above 20%. While some of the results may be explained by poor signal quality for the non-invasive method, we conclude that in healthy dogs ICG-PDR performed poorly.

## 1. Introduction

Indocyanine green (ICG) is an indicator dye that is exclusively cleared from the bloodstream by hepatic uptake and reportedly has a wide therapeutic margin [1,2]. Consequently, plasma clearance of ICG (ICG-CL) has been used in many experimental studies for its potential to detect and stage varying parenchymal and non-parenchymal hepatic diseases in dogs [3,4,5]. However, determining ICG-CL traditionally demands repeated blood sampling to accurately characterize its temporal disposition after intravenous bolus administration [6]. Plasma ICG concentration analysis requires high-performance liquid chromatography (HPLC) [7], which further reduces the clinical practicality of ICG-CL testing of dogs with suspected liver pathologies.

Previously, we reported a good correlation and potential agreement between traditional ICG clearance (ICG-CL) and a non-invasive, transcutaneous method in dogs subjected to isoflurane anesthesia [8]. The latter technology is based on continuous infrared detection of ICG with a near-infrared spectroscopy sensor pre-shaped to be applied on a person’s finger (LiMON module, PulsioFlex, Pulsion Medical Systems SE, Feldkirchen, Germany). Instead of calculating clearance based on ICG concentrations in plasma samples, the monitoring device calculates ICG plasma disappearance rates (ICG-PDR) and retention estimates (ICG-RE) as a percentage change over time [8,9]. In short, to estimate the decay of the initial ICG concentration in the blood normalized to 100%, the method requires the placement of the LiMON probe on the shaved tail and the intravenous administration of a single intravenous ICG bolus. The final transcutaneous test results are automatically recorded by the monitoring platform’s software and stored as a PDF [8]. While the initial correlation between these two methods in dogs is promising and might potentially yield a practical, patient-side diagnostic tool for assessing canine liver function, free of repeated blood sampling and costly concentration analyses, further evaluation of ICG-PDR and PCG-RE is required prior to transferring this method to client-owned dogs with clinical liver disease [8].

In view of that, we aimed to further evaluate the correlation between ICG-CL and ICG-PDR in healthy dogs. We hypothesized that both methods would be able to detect and differentiate between the impact of two sedation protocols known to produce markedly different hemodynamic outcomes in dogs [10,11,12]. Secondly, we aimed to investigate the repeatability of ICG-PDR and ICG-RE in healthy conscious dogs. We hypothesized that the intra-subject variability coefficients from repeated studies would remain low, proving adequate repeatability for a diagnostic test targeted for clinical use.

## 2. Materials and Methods

### 2.1. Animals

Eight healthy purpose-bred Beagles from the Experimental Animal Facilities, University of Helsinki, including 2 females and 6 males, aged between 2 and 3 years, and weighing between 12.6–16.2 kg were used in the study. The dogs were housed according to European Union guidelines (groups in indoor pens with the possibility for outdoor runs). The indoor environmental temperature was maintained at a range of 15–24 °C. The dogs were exposed to both natural and artificial light (between 07:00 and 16:00). They were fed with a standard commercial diet. The food was withheld for 12 h prior to all experiments. Dogs were determined to be healthy based on physical examination, complete blood count and serum biochemistry, including the activities of liver enzymes (alanine aminotransferase and alkaline phosphatase). All studies were conducted at the Veterinary Teaching Hospital, University of Helsinki. The experiment was approved by the Animal Experiment Board in Finland (permit ID ESAVI/7187/04.10.03/2012; approved 5 December 2012).

### 2.2. Study Design

In the first phase, the dogs were randomly allocated to receive one of two sedation protocols in a cross-over design: medetomidine 10 μg/kg (Dorbene 1 mg/mL, Laboratories SYVA, Leon, Spain [Med]) with or without vatinoxan 200 μg/kg (Vetcare Ltd., Helsinki, Finland [MedVat]). All dogs received both treatments as an intravenous bolus at least 14 days apart. Immediately prior to receiving either treatment, ICG-CL, ICG-PDR and ICG-RE were determined as previously described [8]. Briefly, 0.5 mg/kg of reconstituted ICG (Pulsion, Munich, Germany) was intravenously administered via an intravenous 20 G catheter (Terumo, Liege, Belgium) placed in the left of the right cephalic vein and venous blood was collected into heparinized tubes from a central venous catheter (MILA International, Florence, KY, USA) placed in the contralateral jugular vein. Samples (6 mL each) were obtained 1, 5, 10, 15, 30 and 60 min after ICG administration. Simultaneously, ICG-PDR and ICG-RE were determined using the PulsioFlex (Pulsion) monitoring platform, with its probe secured in the clipped ventral surface of the base of the dog’s tail. After obtaining the last blood sample, Med or MedVat was administered, and 20 min later the procedure was repeated. During sedation, indirect systolic blood pressure was intermittently measured with a Doppler unit (Parks Medical, Aloha, OH, USA). After collection of the last blood sample, all dogs received 50 μg/kg of atipamezole (Antisedan, Orion Pharma, Finland) intramuscularly, catheters were removed and dogs were returned to their kennels once they had fully recovered.

In the second phase, ICG-PDR and ICG-RE determinations were repeated three times in six of the dogs, once daily for three consecutive days. No blood samples were collected and all dogs were conscious during the studies.

### 2.3. Plasma ICG Concentration Analysis and Calculations

Care was taken to prevent sample exposure to ambient light. All blood samples were centrifuged at 3000× *g*, 15 min, +4 °C, within 3 h post-collection. Plasma was transferred to cryotubes, frozen and stored at −80 °C. Analysis of the samples was performed within 1 month using an HPLC-UV method described previously [13]. ICG-CL was determined with a non-compartmental model analysis. Since ICG-PDR and ICG-RE are determined from continuous data which the monitoring platform collects for 15 min after ICG administration, ICG-CL was similarly calculated from the plasma concentrations obtained during the first 15 min and thus, concentration data from the 30- and 60-min samples were disregarded. Similarly, areas under the time–concentration curves were calculated for the first 15 min (AUC_0–15_), with ICG concentrations at time 0 min (C_0_) extrapolated from the individual plots. Furthermore, in order to compare retention percentages between the two methods, 15-min retention percentages (ICG-R15) were calculated from the ICG plasma concentration data as C_15_/C_0_ × 100 (%). All pharmacokinetic estimates were calculated with a commercial software package (2023R1) (Monolix Suite, Lixoft, Antony, France).

### 2.4. Statistical Analysis

No ad hoc power analysis was performed per se as the sample size was dictated by the number of available animals. However, based on our previous work we estimated that eight dogs would suffice to reject a null hypothesis of no significant difference in ICG-CL between conscious dogs and dogs sedated with Med. The distribution of the data was examined using a Shapiro-Wilks test. Correlations between ICG-CL, ICG-PDR, ICG-RE and ICG-R15 were estimated with both Spearman’s and Pearson’s tests. Differences between Med, MedVat and conscious states were tested with repeated-measures analysis of variance and post hoc t-tests with Bonferroni corrections. The repeatability of ICG-PDR and ICG-RT was assessed by calculating coefficients of variation (CV%) for the six dogs studied in the second phase as standard deviation/arithmetic mean × 100 (%). Data were computed with a commercial statistical software package (SPSS v28.0, IBM, Armonk, NY, USA) and are presented as the mean ± standard deviation or median (range). The alpha level for significance was set below 0.05.

## 3. Results

Plasma ICG concentrations are presented in Figure 1. The results from a total of 32 determinations of ICG-CL, ICG-PDR, ICG-RE, ICG-R15 and AUC_0–15_ obtained in the first phase are summarized in Table 1. No significant differences were detected in ICG-PDR or ICG-RE between the conscious values and values obtained during Med or MedVat sedation, whereas AUC_0–15_ increased and ICG-CL decreased significantly with Med. Overall, correlation coefficients from conscious dogs for both Pearson’s and Spearman’s correlations analysis between ICG-CL and ICG-PDR remained below 0.5, and for Med yielded weak negative values, whereas for MedVat the Pearson coefficient was significant at 0.75. The associations between the two retention parameters, ICG-RE and ICG-R15, were similarly weak for both coefficients, apart from for MedVat (Table 1). On multiple occasions, the PulsioFlex monitor alarm for poor signal quality appeared during sedation with Med. Both Med and MedVat induced profound sedation in all dogs. All dogs remained normotensive during both sedations and recovered uneventfully.

The results from the second phase are presented in Table 2. Overall, repeatability for both ICG-PDR and ICG-RE was very poor with a CV below 20% in only one dog.

## 4. Discussion

In contrast with our previous results, ICG-PDR did not appear to have a clear association with ICG-CL, the latter of which could be considered as the reference standard for this type of test. While somewhat surprising, there are possible explanations for this discrepancy. First, the signal quality acquired from the transcutaneous probe placed on the tail may have varied more during the present study when compared to dogs under or recovering from general anesthesia with isoflurane [8]. In fact, as the determination of ICG-PDR takes approximately 15 min after ICG administration, it would be very unlikely that an active, healthy dog remains immobile for the entire duration of the measurement. To further complicate issues with signal quality, sedation with Med probably deteriorated the signal even further by inducing systemic vasoconstriction [10]. In contrast, sedation with MedVat probably yielded the ideal conditions for ICG-PDR, since systemic vasoconstriction was not expected, the dogs were profoundly sedated and peripheral perfusion remained relatively unaffected [14]. In accordance, the Pearson correlation between ICG-CL and ICG-PDR was significant only in dogs sedated with MedVat, suggesting a linear relationship between the two parameters. In contrast, the weak negative correlation in dogs sedated with Med and overall, very low values obtained from conscious animals between the two parameters are indicative of poor association between the two methods, which would reflect adversely on the diagnostic precision of ICG-PDR. While a severely ill dog is more likely to remain immobile and less likely to be sedated with medetomidine, only an appropriately designed clinical study in dogs with varying types and degrees of liver disease could confirm whether ICG-PDR would provide any additional benefit to diagnostic tests that are presently utilized to directly or indirectly assess liver function.

The early plasma ICG concentrations were significantly higher with Med than when the dogs were conscious, which appeared to have a large enough impact on both ICG-CL and AUC_0–15_ to result in significant differences in these parameters. In contrast, no significant differences were found with ICG-PDR. This was somewhat surprising, as Med has been reported to drastically reduce cardiac output and hepatic blood flow in dogs [10,12,15]. However, it is likely that the plasma clearance of ICG is limited by the rate of uptake rather than the degree of perfusion [16]. In other words, even if there were differences in hepatic blood flow between the treatments and/or conscious states, in the absence of parenchymal disease minor differences in the resulting functional ICG uptake would remain undetected by ICG-PDR. It is also worth noting that the first samples obtained one minute after ICG administration, were possibly subject to incomplete mixing within the central compartment. While the ICG tests were always performed a full 20 min after Med or MedVat, cardiac index values remaining below the estimated total canine blood volume for longer than half an hour have consistently been repeated for similar intravenous medetomidine doses [10,17]. Thus, it is possible that the early high ICG concentrations observed with Med reflected, at least partially, differences in ICG disposition rather than a true effect on functional hepatic uptake. In retrospect, more frequent early sampling would likely have yielded more accurate clearance estimates. However, omitting the first samples from the analysis had no significant effect on estimates calculated with a non-compartmental pharmacokinetic analysis. Taken together, it remains possible that ICG-PDR would perform differently in detecting hepatic dysfunction due to liver disease in dogs.

No significant difference was found between conscious and sedated dogs for either ICG retention parameters. While the detailed algorithm for ICG-RE is not available, at least the way ICG-R15 was calculated in this study appeared sensitive to the impact of C_o_ estimates which, again, were larger in dogs sedated with Med. Consequently, ICG-CL and ICG-R15 appeared to change in the same direction when the dogs were sedated with Med, which is less likely to be explained by Med-induced changes in ICG uptake. Correlations between the retention parameters were weak at best, again except for values obtained from dogs sedated with medetomidine and vatinoxan, probably due to the combination of cardiovascular effects and the transcutaneous signal quality.

The repeatability of ICG-PDR and ICG-RE was rather poor. While the explanation may be the poor signal quality in healthy conscious animals, CVs consistently above 20% in all but one dog, and a median CV above 40% would make it an unacceptably performing diagnostic test [18]. It would appear to be unlikely that intra-subject functional hepatic uptake of ICG would vary to such an extent, even if intersubject variability for ICG-CL is affected by differences in physiological function [19]. 

This study had some limitations. Our sample size was small and consisted only of dogs of a single canine breed, and with no apparent liver disease, which limits the applicability of our conclusions. In addition, occult liver dysfunction was not completely excluded and the dogs’ hematology and serum biochemistry panels were not tested for potential correlations with ICG clearance or retention parameters. Moreover, we did not measure the effects of the two sedation protocols on cardiac output or hepatic blood flow. It is possible, that contrary to many previous studies, there were no relevant differences in cardiovascular performance between the sedation protocols or conscious states. Nevertheless, the noninvasive, infrared-based method appeared to perform poorly in this study, possibly due in part to signal quality issues, which should be considered when planning future investigations in dogs with clinical liver disease. In conclusion, ICG-PDR should not be adopted in clinical practice without critical evaluation of its accuracy and precision in dogs with pre-existing liver dysfunction.

## Figures and Tables

**Figure 1 animals-13-03455-f001:**
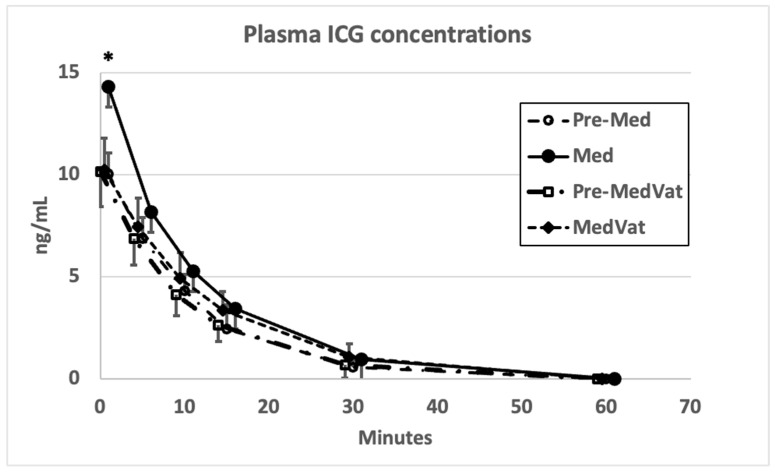
Plasma ICG concentrations in eight dogs while conscious (Pre-Med = prior to medetomidine and Pre-MedVat = prior to medetomidine–vatinoxan) and sedated (Med = after medetomidine and MedVat = after medetomidine–vatinoxan). Error bars indicate standard deviations. * denotes *p* < 0.05 significant difference between Med and Pre-Med.

**Table 1 animals-13-03455-t001:** Results (mean ± standard deviation) for eight dogs and correlation coefficients between ICG-CL and ICG-PDR (r1) and between ICG-RE and ICG-R15 (r2), respectively.

Parameter		All	Conscious	Med	MedVat
N		32	16	8	8
ICG-CL (mL/min/kg)		3.9 ± 1.0	4.4 ± 1.0	3.3 ± 0.4 ^a^	3.8 ± 1.0
ICG-PDR (%/min)		8.2 ± 2.4	9.0 ± 2.2	8.1 ± 3.0	6.7 ± 1.3
ICG-R15 (%)		25.6 ± 7.6	25.2 ± 7.1	22.2 ± 8.5	29.9 ±6.2
ICG-RE (%)		31.2 ± 9.6	27.1 ± 7.6	32.5 ± 10.9	37.5 ± 6.8
AUC_0–15_ (min·ng/mL)		97.6 ± 18.7	87.3 ± 15.9	115.8 ± 7.8 ^a^	97.6 ± 9.8
Pearson	r1	0.31	0.29	−0.11	0.75 ^b^
	r2	0.15	0.11	−0.35	0.68 ^b^
Spearman (r)	r1	0.42	0.26	−0.36	0.62
	r2	0.16	0.14	−0.08	0.68 ^c^

^a^ *p* < 0.05 significantly different from Conscious ^b^ *p* < 0.05 significant linear association ^c^ *p* < 0.05 significant monotonic association.

**Table 2 animals-13-03455-t002:** Day-to-day variation of ICG-PDR and ICG-RT and coefficients of intrasubject variation (CV) in six dogs (data presented as median (range)).

Parameter	Day 1	Day 2	Day 3	CV (%)
ICG-PDR (%/min)	6.7 (4.1–12.3)	8.4 (4.8–15.5)	7.3 (5.1–12.2)	42 (9–53)
ICG-RE (%)	36.7 (15.8–54.1)	28.7 (9.8–48.7)	34.0 (16–48.5)	51 (9–64)

## Data Availability

Data are contained within the article.

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
