# Peer review of "Validity and Repeatability Characteristics of a Non-Invasive, Infrared-Based Method Estimating Plasma Indocyanine Green Decay in Healthy Dogs"

_animals, 2023, doi:10.3390/ani13223455_

Round 1

Reviewer 1 Report

Comments and Suggestions for Authors

General comment for the authors:

The manuscript (original article, brief communication) describes a prospective, cross-over interventional study in healthy research beagle dogs to evaluate the validity and reproducibility of indocyanine green clearance/retention parameters as a prelude to evaluating their potential as a clinical (surrogate) diagnostic test in dogs with a suspicion of hepatic disease/dysfunction.

The study design is sound and well-described, and the conclusions drawn from the results of the study are valid and plausible. A couple of comments warrant to be addressed:

Specific comments

Title page, line 11: Consider changing “liver disease” to “liver dysfunction”.

Title page, line 19 and page 2, line 78: Please include the study design.

Page 2, lines 47-48: This sentence needs to be rephrased, specifically it remains unclear what the term “not unlike” refers to.

Page 2, lines 54ff: Could a portion on potential (systemic and/or organ) toxicity of indocyanine (or lack thereof) please be included? Particularly with potentially delayed clearance/increased retention, the reader should be informed whether toxic effects are a potential concern.

Page 2, line 60: Remove the comma that follows “hypothesized”.

Page 2, line 72: Was a urinalysis/urine sediment analysis done as well in these dogs?

Page 2, line 75 and page 6, line 237: The date of approval (for PH1446A) should be added.

Page 2, line 85: Consider rephrasing “either cephalic vein” as “the left or right cephalic vein”.

Page 4, Table 1: Significant results could be highlighted in bold font for easier appreciation.

Page 4, lines 157-158: It remains unclear why a “20% cut point” is used. This should be supported by a reference and briefly introduced in the introduction and/or materials and methods.

Page 5, line 180: I would suggest replacing “were rather disappointing” with a (brief) explanation of the clinical consequence (e.g., diagnostic utility) of this (disappointing) finding.

Page 5, line 182: Consider replacing “properly conducted” with “appropriately designed”.

Page 5, line 184: Consider replacing “commonplace” with “utilized to directly or indirectly assess liver function”.

Page 5, line 190: Should be “[…] by the rate of uptake […]”.

Page 5, line 191: Should be “[…] than the degree of perfusion […]”.

Page 5, line 193: Consider rewording as “[…] ICG uptake would remain undetected […]”.

Page 5, line 204: Consider rephrasing “All in all” as “Taken together, […]”.

Page 5, line 205: Consider extending to “[…] in detecting hepatic dysfunction due to liver disease […]”.

Page 6, lines 211: Consider rewording as “[…] would less likely be explained by MED-induced changes […]”.

Page 6, lines 213-214: Suggest changing the last part of the sentence to “[…] probably due to the combination of cardiovascular effects and the transcutaneous […]”.

Page 6, line 215: Instead of “far from ideal” it should be “rather poor”.

Page 6, lines 215-2168: Consider changing this paragraph as follows: “While an explanation may be the poor signal quality in […], CVs consistently above 20% in all but one dog would make it an unacceptably performing diagnostic test [18]. It would appear to be unlikely that intra-subject […].”

Page 6, lines 220ff: With a full hematology and serum biochemistry panel available from these dogs, could other parameters (microcytosis; pseudo-functional liver parameters, bile acids, bilirubin, albumin, BUN etc.) be evaluated for a potential correlation with these ICG clearance/retention parameters if occult liver dysfunction/disease cannot be completely ruled out in these dogs?

Page 6, line 222: Consider using more careful wording (e.g., “dogs without apparent liver disease”) as occult hepatic conditions/subclinical hepatic dysfunction cannot be entirely excluded without a more invasive diagnostic work-up. In addition to the small numbers, a limitation of the study would also be that ICG clearance/retention was studied in healthy dogs of a single canine breed. Were these dogs previously utilized for other studies that could potentially have affected the results of this study?

Page 6, lines 228ff: An overall conclusion should be added for the findings of this study or potential outlook for further investigations.

Comments on the Quality of English Language

Good quality. Minor improvements needed and suggested (see specific comments).

Reviewer 2 Report

Comments and Suggestions for Authors

The authors compare the plasma clearance of indocyanine green (ICG-CL) in six - eight healthy dogs with what they call plasma disappearance (PDR) and retention estimates (RE) obtained from an infrared-based tool. They found a poor correlation between the infrared method and the plasma level. A number of reasons and limitations were shared. 

Overall what is presented is well written however there is a general lack of detail throughout the manuscript. Very little information about the infrared technology is shared and nothing about how PDR and RE are determined or calculated. 

More information is needed on the probe that was used with the Pulsioflex. 

The temporal aspects of the infrared technology are also unclear. The authors mention that PRD takes 15 minutes to determine, again it is not clear why that is the case or how PDR is measured.  

It is not clear what time point/interval was used for the data in table 1 and the units between compared parameters appear to be different. 

It is not clear why MED and MEDVAT need to be capitalized. They are not capitalized in the Figure 1 legend, This should be standardized ideally without excessive capitalization for non-acronyms. 

The paper seems to focus more on the anesthesia protocol than the measuring modality however this is given very little attention in the introduction. If the study aim involves testing anesthesia methods that should be further explained earlier in the manuscript. 

Given how little is presented in the earlier sections of the paper it is hard to have significant comments on the discussion. The authors do a good job of describing the many variables and limitations that may have contributed to the results. It would be nice if some of these were tested or addressed prior to submission for publication. 

The authors do not overstate the results of the paper and the conclusions appear supported by the data provided. 

The formatting of reference 19 needs to be revised to address the excessive use of capital letters in the article title in order to be consistent with the other references. 

Round 2

Reviewer 1 Report

Comments and Suggestions for Authors

General comment for the authors:

Thank you for the thorough revisions of the brief communication and careful consideration of my comments and concerns. The quality of the brief communication has improved, but a few points remain to be addressed before the manuscript can be recommended for publication. Please see specific points below.

Specific comments

Page 2, line 89: Please correct “left or right”.

Page 6, lines 223-224: Including the 20% cut-off would still seem of value. Consider extending the current version of this sentence as follows: “[…] conscious animals, CVs consistently above 20% in all but one dog and a median CV >40% would make […].”

Page 6, lines 227ff: Given that full hematology and serum biochemistry panels were not evaluated for a potential correlation with ICG clearance/ retention parameters, occult liver dysfunction/ disease cannot be completely excluded in these dogs and this needs to be added as a limitation of the study.

Page 6, lines 236ff: An overall conclusion and potential outlook for further investigations need to be added.

Reviewer 2 Report

Comments and Suggestions for Authors

Reviewer comments are meant in part to help improve the quality of the final manuscript. Reviewers like myself identify methodological concerns, areas that require improvement, and suggestions. I appreciate when some effort is made to address these rather than refute or ignore them completely. I see that was the case not only for my comments but also those of the other reviewer. When suggestions are not feasible, these can be addressed in study limitations or as future directions. I ask that the authors look at the 1st set of reviewers from both reviewers and attempt to address all of the comments. 

Round 3

Reviewer 2 Report

Comments and Suggestions for Authors

I would to thank the authors for their willingness to revisit and address reviewer comments. I did not catch that the abridged nature of the manuscript was due to the communication designation of the manuscript. That being said the authors have sufficiently addressed most of my concerns. The current revised version is acceptable for publication.